# Brief Sensations Seeking Scale (BSSS): Validity Evidence in Mexican Adolescents

**DOI:** 10.3390/ijerph19137978

**Published:** 2022-06-29

**Authors:** César Merino-Soto, Edwin Salas-Blas, Berenice Pérez-Amezcua, Javier García-Rivas, Omar Israel González Peña, Filiberto Toledano-Toledano

**Affiliations:** 1Instituto de Investigación de Psicología, Universidad de San Martín de Porres, Lima 34, Peru; sikayax@yahoo.com.ar (C.M.-S.); e.salasb@hotmail.com (E.S.-B.); 2Centro de Investigación Transdisciplinar en Psicología, Universidad Autónoma del Estado de Morelos, Pico de Orizaba 1, Los Volcanes, Cuernavaca 62350, Mexico; berenice.perez@uaem.mx; 3Centro Interamericano de Estudios de Seguridad Social, San Ramon S/N, San Jeronimo Lidice, Magdalena Contreras C.P., Mexico City 10100, Mexico; saverio_us@hotmail.com or; 4Institute for the Future of Education, Tecnologico de Monterrey, Av. Eugenio Garza Sada Sur No. 2501, col. Tecnologico, Monterrey 64849, Mexico; ogonzalez.pena@gmail.com; 5Unidad de Investigación en Medicina Basada en Evidencias, Hospital Infantil de México Federico Gómez National Institute of Health, Márquez 162, Doctores, Cuauhtémoc, Mexico City 06720, Mexico; 6Unidad de Investigación Sociomédica, Instituto Nacional de Rehabilitación Luis Guillermo Ibarra Ibarra, Calzada México-Xochimilco 289, Arenal de Guadalupe, Tlalpan, Mexico City 14389, Mexico

**Keywords:** sensation seeking, risk behaviors, validation, measurement invariance, meta-analysis

## Abstract

Sensation seeking is a construct associated with risky behaviors over a wide age range, but validation studies in Mexico are lacking. The aim of this study was to investigate the validity of two versions of the Brief Sensation Seeking Scale (the BSSS-8 and BSSS-4) in young Mexican individuals. The sample consisted of 2884 students (age: M = 16.6, SD = 1.5) from five preparatory schools in Morelos, Mexico. The internal structure of the BSSS was evaluated according to the structural equation modeling (SEM) parameterization, including measurement invariance (compared to the factor loadings obtained in the meta-analysis); conditional reliability; and equivalence between versions. The unidimensionality and measurement invariance (configurational, factor loadings, thresholds, intercepts, and residuals) across sex and age groups were satisfactory, and the factor loadings were highly congruent with those obtained in the meta-analysis. Reliability was suitably high (greater than 0.80), especially near the mean scores, but was lower for extreme scores. Thus, the instrument was concluded to be optimal for defining the construct of sensation seeking, consistent with the findings of previous studies.

## 1. Introduction

Although initially sensation seeking (SS) was considered a biologically based personality trait, research has shown that it is fundamentally influenced by contextual variables. The influence of contextual variables, and thus multicausality, is widely accepted in health disciplines [1,2,3].

SS encompasses the search for risky physical and social experiences that individuals find enjoyable [4]. The associations between SS and public health problems, such as addiction [5,6,7,8,9,10]; antisocial behavior [11,12,13,14]; aggression [15,16]; risky sports [17]; suicide [18]; and sexual behaviors [19], are well established.

SS is linked to alcohol consumption and smoking and is associated with other predictive factors for these behaviors. Specifically, emotional symptoms, including increased symptoms of anxiety and (especially) depression [20], have been found to be physiologically linked with smoking [21] and, therefore, with nicotine consumption [21,22,23]. Because of the covariation of SS and emotional symptoms, the inclusion of SS measures in mass screening or screening assessments in adolescents may be useful. However, including measures of SS would entail evaluating the cost-effectiveness of these assessments.

For decades, SS has been measured by the Sensation Seeking Scale V (SSS-V; [24]), which has been validated in numerous cultures, languages, and populations (e.g., recently in Manna et al. [25]). The SSS-V showed some problems that limited its use in the following years. These limitations were the extension with respect to the number of items, the unstable relationship of the items with their intended constructs to measure, the age of the content, the format of items in dichotomous responses, and the reference to adult situations [26,27]. The demand for a brief, adolescent-usable, unidimensional, and group-invariant measure motivated the development of a measure, the Brief Sensations Seeking Scale (BSSS; Hoyle et al. [26]). The BSSS is a shorter version developed to increase intercultural generalization and parsimony and reduce dimensionality. To date, there has been a comparative increase in the number of BSSS studies compared to SSS-V, and this trend is likely to continue for many years. However, recent studies (presented in Table 1) suggest that several aspects of the BSSS are not addressed or inconsistently addressed in the analyses of its internal structure. These aspects include the absence of conditional reliability, the equivalence of the full and short versions, treatment of items as continuous variables, psychometric equivalence between groups, and the fact that internal consistency is mainly estimated by the *α* coefficient. Furthermore, the psychometric value of the abbreviated 4-item version (the BSSS-4; [27]) remains unclear.

In sum, recent studies have indicated that (a) the conditional reliability of the BSSS may represent a realistic characteristic of score accuracy at different score levels [33]; (b) the equivalence of the two versions of the BSSS is required to guarantee the conceptual representativeness of a modified version of the instrument and establish consistency in decisions using the scores [34,35]; (c) treating categorical items as continuous variables produce spurious and unrepresentative estimates of their statistical properties [36]; (d) the coefficient *α* is not an efficient measure of internally consistent if there are correlated errors and wide-ranging differences in factor loadings [37]; and (e) measurement invariance is an aspect of internal structure that allows the use of the instrument to compare groups, such as with means or correlations, but it is infrequently assessed in BSSS [33].

Although some aspects of validity can be inferred, these inferences are approximations without an empirical basis according to our data [38,39], and empirical verification of the properties of the BSSS is good practice, as it ensures correct interpretation of its scores [33]. Given the existing abbreviated version of the BSSS (i.e., the BSSS-4), it is surprising that only a single study has attempted to evaluate its psychometric properties, especially equivalence with the full version (i.e., the BSSS-8). The brevity of the BSSS-4 suggests that it is optimal for mass screening evaluations as it does not require a substantial time investment to complete. Although, in an absolute view, the time to complete the BSSS-8 version does not differ substantially from the time for the BSSS-4, from a relative angle, this difference may be an influential factor. Specifically, this time difference in the context of a long survey and administered under moderately optimal conditions may not be perceived as trivial. Additionally, in an online survey where direct monitoring may not be in the control of the researcher, the brevity of the survey will be maximized with few items but without losing validity, and especially in sensation seeking measurement [26,27,31].

However, the modifications of the BSSS-4, which aimed to reduce the number of items, may produce different results than the original instrument [40,41], and the benefits of a shorter instrument do not justify inferring that its psychometric properties are identical to the full version. In this sense, one possible misuse is to induce the validity of the BSSS-4 version using the validity evidence of the BSSS-8 [38,39], and it is required to assess the validity of the BSSS-4 due to the small number of studies.

As seen in Table 1, no study employing the BSSS has been conducted in Mexico, and the only relevant Hispanic American studies took place in Spain [32] and Peru [29,31]; thus excluding Mexican samples. The motivating factors for designing the BSSS-4 are its brevity, parsimony, reduced dimensionality, and potential usefulness in epidemiological studies, all of which are beneficial. Thus, the present study aimed to evaluate its internal structure and its validity in relation to other variables in young and adolescent Mexican participants.

This study investigated the internal structure of the BSSS-4, including the dimensionality, measurement invariance, and internal consistency; these sources of validity are necessary for defining the construct and the algorithm to obtain the BSSS score [33]. In this factor loading-based psychometric meta-analysis, based on the literature reviewed in Table 1, we hypothesized that the items in both versions of the BSSS represent a unidimensional construct.

Regarding its validity with other variables, different versions of the BSSS should be established by creating and adapting measures of related attributes (e.g., depressive symptoms and tobacco use, which have an established relationship with SS; [22,23,42]), which will provide an opportunity to evaluate the discriminative validity of sensation seeking measures such as the BSSS. Thus, we hypothesized that scores on both versions of the BSSS would be positively associated with depressive symptoms and alcohol and tobacco use.

Finally, the present study also examined the incremental validity of the BSSS, with depressive symptoms as a competing variable. Incremental validity can indicate utility in symptom differentiation [43,44,45], specifically, the instrument’s added value in clinical assessments. Since the incremental validity of the BSSS has not been reported in previous studies, we also aimed to bridge this gap. We hypothesized that scores on both versions of the BSS would demonstrate incremental validity for detecting people who use alcohol and tobacco beyond the use of depressive symptoms.

## 2. Materials and Methods

### 2.1. Study Design

This study had a cross-sectional and instrumental design [46,47].

### 2.2. Participants

Student participants from 5 high schools in the state of Morelos (Mexico) were selected. These schools were located in the following cities: Cuernavaca, Jiutepec, Tepoztlán, and Cuautla (2 schools were located in this city). The distribution of the total sample (3007 students) across these 5 schools was 906 (31.4%), 362 (12.6%), 785 (27.2%), 328 (11.4%), and 503 (17.4%), respectively. The inclusion criteria were the ability to provide voluntary informed consent and enrollment on the corresponding campus. Cases that included data with multivariate outliers were excluded (see 2.6. Statistical Analysis).

### 2.3. Measures

#### 2.3.1. Brief Sensation Seeking Scale (BSSS)

This scale was a single dimension sensation seeking instrument applicable to adolescents and adults. We used the 8-item (BSSS-8; [26]) and 4-item (BSSS-4; [27]) Spanish versions [31]. In these scales, answers to each item were ordinally ranked with 5-level points (from “strongly disagree” to “strongly agree”. In previous studies (Table 1), the reliability of this scale was reported as higher than 0.70, and the unidimensionality was confirmed.

#### 2.3.2. Center for Epidemiological Studies Depression Scale-7 (CESD-7)

This 7-item self-report questionnaire was used to assess the depressive symptoms of dysphoric mood and reduced motivation, concentration, pleasure, and sleep. We used the version adapted for Mexico [48], which was derived from the original version by Santor and Coyne [49]. All items, except for item 6, were oriented toward depression, and 4 ordinal response options (from “never” to “always”) were provided. In the present study, the unidimensionality of the CESD-7 was satisfactory (WLSMV-*χ*^2^ = 171.61, df = 9, *p* < 0.01; CFI = 0.989, SRMR = 0.05, WRMR = 2.28; factor loadings between 0.19 and 0.81). Item 6 was not included due to its very low factor loading (at 0.02). The internal consistency of the 6 items was *ω* = 0.85 (95% CI = 0.84, 86; se = 0.005).

#### 2.3.3. Alcohol and Tobacco Use

Two binary response (yes or no) items were created to measure alcohol and tobacco use. The instructions were as follows: “Next, we ask that you provide information about your consumption of various substances. Please answer as honestly as possible and remember that your answers are confidential. Have you ever used the following substances? If so, please answer ‘Yes’. If you have never tried a given substance, please answer ‘No’.” The specific items for alcohol and tobacco use were presented sequentially.

#### 2.3.4. Sociodemographic Information

An additional sheet contained questions about participant age, sex, semester of study (semesters are the number of semesters completed by students, and it is an indicator of their progress in their studies), marital status, and other demographic variables.

### 2.4. Ethical Considerations

This study was a part of the research project HIM/2015/017/SSA.1207: “Effects of mindfulness training on psychological distress and quality of life of the family caregiver”, which was approved on 16 December 2014, by the Research, Ethics, and Biosafety Commissions of the Hospital Infantil de México Federico Gómez, National Institute of Health, in Mexico City. This study was conducted in accordance with the ethical rules and considerations for human research currently recommended in Mexico [50] and those outlined by the American Psychological Association [51]. All family caregivers were informed of the objectives and scope of the research and their rights according to the Declaration of Helsinki [52]. The caregivers who agreed to participate in the study signed an informed consent letter. Participation in this study was voluntary and did not involve payment.

### 2.5. Procedures

The questionnaire was electronically administered between the months of April and May 2019 with the supervision and support of counselors on each campus. They received training from the research group to resolve any doubts and discuss other possible issues. Instruments were presented in the same order at all schools (i.e., informed consent, demographic questions, and instrument questions), and the administration procedure was held constant. First, the consent form was given to the parents or guardians; thus they could decide whether to allow their children to participate in the study. Second, informed consent was provided by each student. Finally, the participants collectively answered the instruments in the computer center of each school.

For ethical considerations, the entire data collection procedure was aligned with the principles of the Helsinki Declaration and the Belmont Report regarding voluntary participation, freedom to withdraw, the anonymity of response, and confidentiality of information. To ensure respondents’ anonymity, no identifying information was collected from the participants, nor was any compensation awarded for participation. Participants were informed of their right to continue or revoke their participation at any point.

### 2.6. Statistical Analysis

To control for possible response biases associated with insufficient effort or random responses, multivariate outliers were detected using the Mahalanobis *D*^2^ distance [53]. Descriptive and distributional statistics were obtained for the items; in addition, associations with age (biserial correlation by points) and sex (correlation by Glass ranks). The R packages *careless* [54], *rcompanion* [55], and *MVN* [56] were used in the analysis.

Analysis of the internal structure was performed using confirmatory factor analysis (CFA-SEM) to evaluate 2 models: (a) the one-dimensional model, specifying a congeneric one-dimensional submodel (i.e., free variability of factor loadings), and (b) the tau-equivalent one-dimensional submodel (i.e., equality of factor loadings). The mean and variance-adjusted unweighted least squares (ULSMV) estimator was selected because it produces a better nonspurious fit compared to other estimates for categorical variables, e.g., the mean- and variance-adjusted weighted least squares (WLSMV) estimator [57]. The fit was evaluated with two main approximate fit indices [58]: the comparative fit index (CFI; good fit, ≥0.95 [58]) and the standardized root mean squared residual (SRMR; good fit, ≤0.05 [58]) index. The root mean square error of approximation (RMSEA) and the weighted root mean square residual (WRMR) were rejected due to the higher rate of false positives when the mismatch was small, the sample was large, and when the data had approximately 5 ordinal categories [58,59]. To estimate point reliability, *α* and *ω* coefficients were calculated, and to determine the conditional precision of the items and the total score, item response theory parameters derived from the results of the CFA-SEM [60] were obtained.

The equivalence of the internal structure was assessed at 2 levels: the first level was evaluated in the present sample (i.e., sex and age) through the measurement invariance approach and following the usual sequential steps [37]: configurational invariance, metric, and thresholds. At the second level, invariance (i.e., equality of factor loadings) was evaluated in comparison to the studies in Table 1 via a meta-analysis of factor loadings with the direct procedure [61,62,63]. According to the Gnambs and Staufenbiel method, a Procrustean rotation was applied to the factor loading matrix obtained in our total sample toward a target matrix constructed from the mean factor loadings based on the studies in Table 1. To obtain the target matrix, the factor loadings of the studies in Table 1 were rotated toward a factor loading pattern that consisted of a 0.60 value for each of the items of the target matrix [62]. This target factor load was considered representative and closely approximated the loadings of the studies cited in Table 1 upon visual inspection. Subsequently, congruence was calculated to evaluate the similarity of the obtained measures (i.e., congruence coefficient *φ*; [64]). Next, a simple average was obtained from the rotated loadings adjusted by the Procrustes rotation (i.e., direct method; [62]).

The equivalence between the BSSS-8 and BSSS-4 was evaluated by (a) a corrected correlation [65] and (b) an ordinal gamma association between the quartile classification of the participants obtained from both versions. In this equivalence assessment, the scores from both versions (full vs. abbreviated) were evaluated using a graphical approach that also explored the degree of localized bias in the scores from both versions [66]. The Bland–Altman method [67,68], which plots the difference between the scores on the *y*-axis and the joint mean on the *x*-axis, was used, including both the raw scores and their equivalent z scores.

Finally, evidence of the instruments’ validity with other variables was assessed with Pearson linear correlation and Spearman monotonic correlation analyses (for both zero-order and partial, controlling for sex and age) of the BSSS-8 and BSSS-4 scores with those of depressive symptoms (CESD scores). Because the 2 types of correlations were sensitive to different forms of dependence between variables [69,70], both were employed. Evidence of incremental validity was assessed by multiple logistic regression to estimate the ability of the BSSS to predict the likelihood of alcohol use and tobacco, controlling for the effect of sex, age, and depressive symptoms. Logistic regression was performed using a hierarchical approach [43,44]: in the first step, the sex, age, and depressive symptoms variables were entered, and in the second step, the BSSS score was entered. The criteria for incremental validity of the BSSS was set at a minimum difference (∆_R_) between McFadden’s *R* of both models in the range of 0.15–0.20 [44]. R was derived directly from the square root of McFadden’s *R*^2^ [44]. The significance of the BSSS score in the last step of the hierarchical logistic regression was estimated using the absolute value of the statistical test for each model parameter [71].

The programs used to analyze internal structure were the R packages *lavaan* [72], *semtools* [73], *psych* [74], *coefficientalpha* [75], and the SPSS *Procrustes syntax* [76]; that used to analyze equivalence objective was *BlandAltmanLeh* [77]; and those used to analyze validity in relation to other variables were *caret* [71], *pscl* [78], and *glm* from the *stats* package [79].

## 3. Results

### 3.1. Sample

A total of 123 participants were detected as multivariate outliers, with *D^2^* at the 4.482 (*p* < 0.05) threshold, and subsequently removed. The effective sample size was thus 2884. There were 1450 men and 1434 women (49.7%) distributed in the following semesters: 2nd (1122, 38.9%), 4th (901, 31.2%) and 6th (861, 29.9%). Of the sample, 99% were between 14 and 20 years old (median: 17 years old; interquatile range (IQR) = 1).

### 3.2. Item Analysis

The set of items was not normally distributed at the multivariate (Henze–Zirkler—*Z* = 24.83284; *p* < 0.01) or univariate levels (Cramer von Mises test > 13.0, *p* < 0.01). Both, Mardia’s skewness (b1p = 4.85, z = 2331.9, *p* < 0.01) and kurtosis (b2p = 88.99, z = 19.08, *p* < 0.01) tests showed non normality. The mean response (approximately 3.0), variability (SD = 1.3), skewness and distributional kurtosis were approximately similar (Table 2). Item correlations with sex and age (Table 2) were essentially zero, although some were statistically significant.

### 3.3. Dimensionality

#### 3.3.1. BSSS-8

In the SEM modeling, the fit obtained was ULSMV-*χ*^2^ = 241.091 (20), CFI = 0.992, SRMR = 0.047. The errors from items 1 and 2 were correlated error (expected standardized parameter, 95% CI = 0.12, 0.19); however, this did not produce substantial variations in the practical fit indices, thus it was not included. Factor loadings were high, except for that of item 4. The tau-equivalent model (equality of factor loadings) was moderately satisfactory: ULSMV-*χ*^2^ = 672.553 (27), CFI = 0.976, SRMR = 0.079. In the item response theory (IRT) estimation (Table 3), the discrimination parameter was high (*a*_irt_ > 1.5) and moderately high for item 4, in which its information function was low at the different attribute levels (from −3.0 to + 3.0).

#### 3.3.2. BSSS-4

The fit obtained was ULSMV-*χ^2^* = 88.520 (20), CFI = 0.988, SRMR = 0.054. Four correlated errors were found (between items 1 and 2, 1 and 8, 2 and 7, and 7 and 8), but they were not included due to the trivial change in the fit indices. Table 3 shows the results of the adjustment, in which the factor loadings remain high (>0.70). In the IRT estimation, the same pattern obtained by the information function was observed for the levels of the latent attribute *θ*.

### 3.4. Measurement Invariance

Measurement invariance was investigated with regard to sex and age (divided into 14–16 years old and 17 years old and over groups). This partitioning in variable age was decided on a basis of convenience for the analyses, thus balancing the size of the samples compared. The equivalence of each item parameter was confirmed in both versions (Table 4; configuration, factor loadings, intercepts, and residuals). The measurement invariance based on the studies reported in Table 1, the Procrustean adjustment to factor loadings, was considered satisfactory because the studies produced high congruence indices (*φ* ≥ 0.98 for the total solution and for each item). The average loadings of factors from the meta-analysis were calculated as 0.62, 0.53, 0.61, 0.57, 0.65, 0.66, 0.59, and 0.67, respectively, and were compared with the factor loadings shown in Table 3 (column heading F). Very high *φ* coefficients were obtained for the one-dimensional solution (*φ* = 0.99) and for the items (*φ* > 0.98), indicating satisfactory measurement invariance with the factor loadings obtained in the meta-analysis.

### 3.5. Internal Consistency

The *α* and *ω* coefficients were similar for the BSSS-8 and BSSS-4 (see Table 3), and for pragmatic use, they can be seen as equals. Within the IRT framework (Table 3: Information, SEM, Stand. Inf.), the information function for the items and scores was slightly positively asymmetric, which indicates that greater differentiation and fewer errors were obtained around the mean level of the score. At the extremes of this distribution, especially at very high scores, the information function was poor. Based on this parameter, the reliability of the BSSS-8 reached coefficients greater than 0.80 between attribute levels −1 and +1. This pattern was similar to that of the BSSS-4 scores, although its conditional coefficients were approximately 0.75. In addition to these results, the confidence intervals of the reliability of the BSSS-4 scores did not overlap with those of the BSSS-8; the internal consistency reduction in the BSSS-4 was significantly lower but still greater than 0.80.

### 3.6. Equivalence between the BSSS-8 and BSSS-4

The uncorrected Pearson linear association between the BSSS versions was very strong (*r* = 0.95, *p* < 0.01), and the corrected linear correlation between the two versions was also high (*r* = 0.86, *p* < 0.01). The ordinal association between the classification in quartiles obtained from the BSSS-8 and BSSS-4 was high as well (*gamma* = 0.96, 95% CI = 0.96, 0.97). These results suggest very high levels of agreement and equivalence for the ranking of respondents on their level in the measured attribute, using either raw scores or quartile ranking.

When evaluating possible anomalies in the equivalence of both versions of the BSSS (BSSS8 and BSSS4) according to the means of the Bland–Altman plots (Figure 1), the following was found: the mean difference in z scores was 2.14474 × 10^−16^ (in raw scores = 12.39), and the corresponding standardized lower and upper limits were −0.60 (raw scores = 3.93) and 0.60 (raw scores = 21.15), with a critical standardized difference of |0.60| (raw scores = 8.76). Figure 1 shows that the standardized score differences were small and did not exceed ±1 SD but were approximately ±0.05 SD. In the lower panel of Figure 1 (differences in the raw scores), the progressive increase in differences below the mean difference and toward differences above the mean difference indicates the linear correspondence of the raw scores and the differences between them.

### 3.7. Association with Other Variables

The monotonic association between BSSS scores and depressive symptoms (CESD score) was barely distinguishable (∆ = 0.006) from the calculated linear correlations (Table 5). Thus, the BSSS-8 and BSSS-4 relationships can be considered equal in practical terms. The size of the association was in the small to moderate range (r between 0.20 and 0.30; [80]).

### 3.8. Incremental Validity

The results are in Table 6. In the BSSS-8, the difference between McFadden’s *R^2^* of the compared models (without and with the BSSS) was statistically significant with respect to alcohol (∆χ^2^ = 63.83, gl = 1, *p* < 0.01) and tobacco consumption (∆χ^2^ = 71.20, gl = 1, *p* < 0.01). The differences between the square root of McFadden’s *R* were 0.057 and 0.033 for alcohol and tobacco consumption, respectively; both values were below the chosen criterion for incremental validity (between 0.15 and 0.20; [44]). Finally, the relative importance of BSSS was approximately 9.0% for alcohol and tobacco use. Compared to the depressive symptoms (CESD scores) included in step 2, the significance of both variables was similar for alcohol (CESD significance = 9.54%, OR = 1.08) and tobacco (CESD significance = 8.48%, OR = 1.08) consumption.

In the BSSS-4, the difference between McFadden’s *R*^2^ was statistically significant for alcohol (∆χ^2^ = 61.95, gl = 1, *p* < 0.01; ∆*_R_*^2^ = 0.016) and tobacco consumption (∆χ^2^ = 58.66, gl = 1, *p* < 0.01; ∆*_R_*^2^ = 0.017). These values were below the chosen criterion for incremental validity (between 0.15 and 0.20; [44]). Finally, the relative importance of BSSS was approximately 7.5% for alcohol and tobacco use. Compared to the depressive symptoms (CESD scores) included in step 2, the significance of both variables was similar for alcohol (CESD relative importance = 9.79%, OR = 1.08) and tobacco (CESD significance = 8.65%, OR = 1.08) consumption.

## 4. Discussion

This study was motivated by the absence of evidence on the internal structure of an instrument measuring SS (the BSSS) in Mexico and evaluated the properties of the BSSS in ways that most previous studies did not address. First, the performance was found to be essentially similar according to all descriptive and distribution indicators. However, the size of the correlations between score, sex, and age was approximately zero, suggesting only a small moderating effect. This trend was similar to that found in other studies (e.g., [28,29,30]).

For both versions of the BSS, the total score interpretation was guaranteed and the hypothesis of the unidimensionality was supported. This result has theoretical implications because SS is a construct comprised of several relatively independent constructs, which has been confirmed by recent studies administering the SSS-V to Latino adolescents [81] and other cultural groups (e.g., Italian adolescents, [25]). However, SS can be conceptualized as a unitary concept, operationally defined with selected behaviors to ensure cross-cultural generalizability, that produces a single score, such as the BSSS. The measurement invariance with respect to previous studies suggests that the items are psychometrically generalizable in cultures with different idiomatic expressions (e.g., Italian: [28]; Portuguese: [30]; Spanish from Spain: Martín et al. [32]; Spanish from Peru: Merino et al. [31], and Romero et al. [29]) and that the meaning of these items are understood among these groups. Therefore, these items seem to represent commonly understood behaviors and define SS in various cultures. We cannot extrapolate beyond this finding because our results were mainly focused on a Mexican sample and a single state within Mexico. Additionally, the internal structure remained unchanged in all the tested restrictions (e.g., sex and age). This is a particular strength of the instrument because it guarantees that the means, variances, and covariances in various groups are easily comparable.

Moreover, the equivalence between the two versions (BSSS-8 and BSSS-4) was satisfactory. Thus, our findings were consistent with those of the Peruvian study [31], which was the only study to evaluate the equivalence between the two versions and found acceptable equivalence for the items and group classification. The differences between the two versions were proportional to the score, which was expected given the strong linear relationship between scores on the BSSS-8 and BSSS-4. These differences were approximately constant across scores, not exceeding a difference above 2 SDs. Thus, the use of the BSSS-4 is optimal for classifying adolescents on SS, especially when an abbreviated version is required.

The reliability of both versions is adequate given the purpose of the instrument: amplitude of the measured construct and reduced number of items. However, precision is more appropriate at medium levels of SS, while at the lowest and highest scores of SS, precision is poor. This implies that additional measures are required for adolescents with low or high scores to corroborate their score in the construct, as well as retesting on the instrument. This limitation in the accuracy of the scores is part of an additional limitation: the internal consistency of the BSSS-4 was lower than that of the BSSS-8 because the confidence intervals of both coefficients did not overlap. This loss of precision is not uncommon in reduced versions of self-report measures [37] and is not necessarily a limitation in the absolute sense but rather in a relative sense. The levels of reliability indicated for BSSS-4 scores are sufficient for group description and for basic research. Even broad quartile-based classifications are possible because of the high ordinal association with the quartile classification of the BSSS-8 scores. Finally, the equality of the *α* and *ω* reliability coefficients has two implications: first, the *α* coefficient may be sufficient to estimate the internal consistency of the scores and sophisticated modeling is not required; and second, item variability on validity (i.e., factor loadings) had no observable effect on the *α* coefficient. This equality of coefficients corroborates the results obtained by previous studies [28,31,32].

The incremental validity of both versions of the BSSS was assessed with a hierarchical logistic regression strategy: the first step included sex, age, and depressive symptom scores (CESD-7) and the second step included the BSSS scores. Upon quantitative assessment with the McFadden R^2^ difference, the BSSS was not strong enough (McFadden *R* < 0.15) to support the incremental value of using the BSSS rather than a measure of depressive symptoms. Therefore, the hypothesis that the BSSS had incremental validity was not supported. This may, however, not be a limitation to using the BSSS as an indicator of SS but an additional criterion when depressive symptoms are assessed for a specific purpose, such as the estimation of alcohol and tobacco use. In assessments that include both the BSSS and a measure of depressive symptoms (specifically, the CESD-7), the BSSS remains an important variable because of the variance it explains when assessing alcohol use and depression indicates that it is a close second to depressive symptom scores in terms of predictive value. An apparent implication is that, with respect to alcohol and tobacco use, both measures of depressive symptoms and SS are interesting and necessary elements of an adolescent screening assessment because the constructs are not interchangeable and have good and clear discriminative validity (i.e., because of their relatively weak relationship). Given these relationships between the scores of both versions of the BSSS and the external variables of interest, the hypothesis of association was supported.

This study had three main limitations. First, the evaluation of invariance regarding other samples (i.e., studies listed in Table 1) was conducted through secondary analysis of the factor loadings, i.e., a meta-analysis of their factor loadings. Although this meta-analytic approach is novel, it only permitted the evaluation of measurement invariance; no other levels of invariance could be tested without complete data (e.g., standard errors and correlation matrices). Consequently, full measurement invariance in a cross-cultural context is still needed to draw appropriate inferences about equivalence between international samples. Second, the predictive value of the BSSS for behavioral outcomes (e.g., smoking or alcohol consumption) and other constructs (e.g., impulsivity) must be assessed because our study was primarily intended to study the full internal structure (i.e., dimensionality, measurement invariance, and reliability). Finally, the possible variability of psychometric properties across schools was not explored, and this may require a multilevel approach. However, because sensation seeking is understood primarily as an individual attribute, it may be reasonable to assume that this variability is not substantial.

Implications of clinical value can be concluded in our study. In this study, we extend the validity of the BSSS in a new population context (Mexico), but we also add other properties not addressed in previous studies that have potential clinical value. For example, item scaling performance is not only a matter of statistical item distribution but is also potentially linked to item clinical validity because the response options have clinical value when it helps differentiate individual risk factors. Second, conditional reliability helps to assess the accuracy of the scores at the construct levels where the interventions are applied, i.e., high levels. Finally, an assessment of a number of individual variables associated with risk behaviors requires the school team to plan the selection of instruments and balance the costs of their administration. In this sense, the BSSS-4 can be an excellent tool for screening variables of clinical impact, especially when it comes to preventing substance use in adolescents. Due to its brevity and consistently satisfactory psychometric properties, the BSSS can be incorporated into a universal screening of adolescents in school. Measures of depression may also be included in this screening, and possibly the instrument used here, the Center for Epidemiological Studies Depression Scale-7 (CESD-7; [48]) may be of particular importance.

Within a psychometric research context of the BSSS, future studies can focus on cross-cultural studies through a complete analysis of measurement invariance, as well as meta-analytic research to synthesize item parameters such as factor loadings, intercepts, and residuals. SEM meta-analysis (MASEM) is indicated as an analysis strategy to synthesize cumulative evidence from independent studies. To date, we can assume that there is enough to meta-analyze the internal structure of the BSS.

## 5. Conclusions

We conclude that the internal structure of both versions of the BSSS (the 8-item and 4-item versions) is highly satisfactory given the robustness of the results, the sample size of this study, and the internal (i.e., age and sex) and external (i.e., previous studies) invariability of its parameters. The results point to the psychometric quality of the instrument and to solid interpretations of its measured construct (i.e., SS). Finally, due to the similarity of the factor loadings obtained in this study and the synthesized factor loadings from previous studies, the internal structural properties of both versions can be inferred with respect to the validity of the items from an emic perspective. Because of the satisfactory internal structure and equivalence, both versions of this scale (the BSSS-8 and BSSS-4) are therefore useful tools for national surveys and cross-cultural studies.

## Figures and Tables

**Figure 1 ijerph-19-07978-f001:**
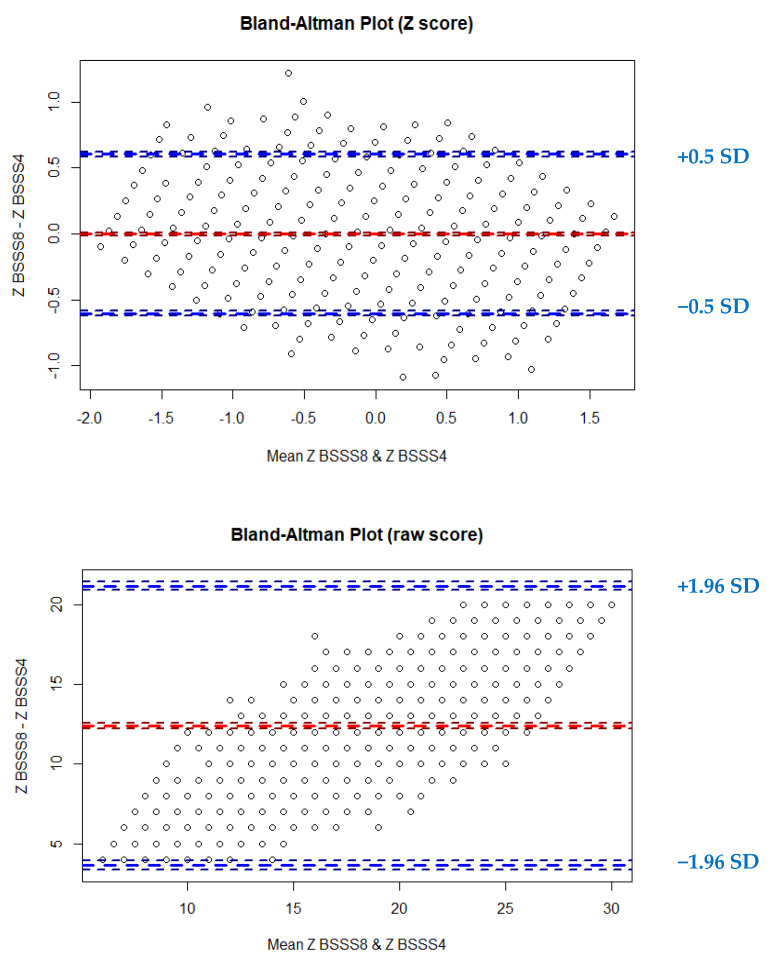
Bland–Altman plot of the z scores and raw scores of the BSSS8 and BSSS4. Z(BSSS8): Z score of the observed raw score on the BSSS-8. Z(BSSS4): Z score of observed raw score on the BSSS-4.

**Table 1 ijerph-19-07978-t001:** Methodology of recent studies on the internal structure of the BSSS (adolescent samples).

Study	Country (*n*)	Reliability	Estimator	R Matrix	Factor Loading	Models	Correlated Errors	Measurement Invariance	Short Version	Equivalence
Primi et al. [28]	Italy (964)	*α* = 0.73 (*ω* = 0.72)	SB-*χ*^2^	N.R.	Min = 0.33 Max = 0.61 Mean = 0.49	Unidimensional Congeneric: Yes Tau-equivalent: N.R.	Item pairs: 1–5 4–8	Sex Age	N.R.	N.R.
Romero et al. [29]	Peru (1003)	*α* = N.R. (*ω* = 0.82)	SB-*χ*^2^	N.R.	Min = 0.46 Max = 0.80 Mean = 0.62	Unidimensional Congeneric: Yes Tau-equivalent: N.R.	Item pairs: 1–2 7–8	Sex Age	N.R.	N.R.
Pechorro et al. [30]	Portugal (412)	*α* = N.R. (*ω* = 0.82)	SB-*χ*^2^	Polychoric	Min = 0.51 Max = 0.87 Mean = 0.69	Unidimensional Congeneric: Yes Tau-equivalent: N.R.	Item pairs: 2–7	Sex Age	N.R.	N.R.
Merino et al. [31]	Peru (618)	*α* = 0.74 (*ω* = 0.74)	SB-*χ*^2^	N.R.	Min = 0.36 Max = 0.64 Mean = 0.51	Unidimensional Congeneric: Yes Tau-equivalent: Yes	N.R.	Sex Age	Yes	Yes
Martín et al. [32]	Spain (1184)	*α* = 0.89 (*ω* = 0.88)	SB-*χ*^2^	N.R.	Min = 0.61 Max = 0.80 Mean = 0.69	Unidimensional Congeneric: Yes Tau-equivalent: N.R.	N.R.	Sex Age	N.R.	N.R.

Note. N.R.: not reported; *α* and *ω*: omega and alpha reliability coefficients; values in parentheses were calculated in the present study from the reported factor loadings. SB-*χ*^2^: Satorra–Bentler modified chi-squared statistic.

**Table 2 ijerph-19-07978-t002:** Results of the univariate and correlation analyses on the BSSS items.

	M	SD	Sk.	Ku.	CVM	Sex (95% CI)	Age (95% CI)
bsss1 ^a^	3.74	1.37	−0.89	−0.48	33.79	0.03 (−0.00, 0.08)	0.03 (−0.07, 0.06)
bsss2 ^a^	2.93	1.39	0.01	−1.25	13.88	−0.07 (−0.10, 0.03)	0.06 * (0.02, 0.09)
bsss3	2.76	1.41	0.17	−1.23	15.28	−0.03 (−0.07, 0.00)	0.02 (−0.00, 0.06)
bsss4	2.60	1.38	0.32	−1.16	17.16	−0.04 (−0.08, −0.00)	0.04 * (0.00, 0.07)
bsss5	3.44	1.47	−0.47	−1.19	23.11	0.07 * (0.03, 0.11)	−0.05 (0.09, 0.02)
bsss6	3.57	1.51	−0.64	−1.07	32.04	0.02 (−0.01, 0.06)	−0.01 (−0.05, 0.02)
bsss7 ^a^	3.05	1.44	−0.06	−1.29	14.30	−0.05 (−0.09, −0.01)	0.05 * (0.01, 0.08)
bsss8 ^a^	2.87	1.35	0.08	−1.10	14.12	−0.01 (−0.05, 0.025)	0.01 (−0.02, 0.04)

Note. bsss: item of the BSSS. Sk: skew coefficient. Ku: kurtosis coefficient. CVM: Cramer von Mises normality test. ^a^ Items of the short version (BSSS-4). * *p* < 0.05

**Table 3 ijerph-19-07978-t003:** Results of the factor analysis and item response theory (IRT) analysis.

	F	BSSS-8 IRT Parameters								F	BSSS-4 IRT Parameters							
		*a* _irt_	Information Function of Latent Attribute Levels (*θ*)								*a* _irt_	Information Function of Latent Attribute Levels (*θ*)						
			−3	−2	−1	0	1	2	3			−3	−2	−1	0	1	2	3
bsss1	0.76	1.96	0.45	0.88	1.11	0.94	0.57	0.26	0.09	0.73	1.24	0.42	0.79	1.01	0.87	0.54	0.26	0.10
bsss2	0.74	1.99	0.22	0.53	0.89	1.04	0.91	0.58	0.26	0.77	1.31	0.22	0.54	0.93	1.10	0.95	0.60	0.26
bsss3	0.71	1.82	0.18	0.42	0.74	0.93	0.86	0.58	0.29	-	-	-	-	-	-	-	-	-
bsss4	0.59	1.31	0.17	0.30	0.44	0.54	0.52	0.41	0.26	-	-	-	-	-	-	-	-	-
bsss5	0.80	2.54	0.39	1.05	1.55	1.43	0.93	0.36	0.10	-	-	-	-	-	-	-	-	-
bsss6	0.74	2.03	0.36	0.82	1.21	1.12	0.68	0.29	0.10	-	-	-	-	-	-	-	-	-
bsss7	0.84	2.81	0.28	0.90	1.45	1.49	1.36	0.79	0.23	0.83	1.41	0.25	0.76	1.31	1.43	1.23	0.66	0.21
bsss8	0.83	2.74	0.24	0.78	1.28	1.25	1.29	1.01	0.40	0.80	1.36	0.21	0.60	1.05	1.18	1.11	0.76	0.32
IRT parameters																		
Information	-	-	2.30	5.68	8.67	8.73	7.11	4.28	1.73	-	-	1.10	2.68	4.30	4.58	3.84	2.28	0.88
SEM	-	-	0.66	0.42	0.34	0.34	0.37	0.48	0.76	-	-	0.95	0.61	0.48	0.47	0.51	0.66	1.06
Stand. Inf.	-	-	0.56	0.82	0.88	0.89	0.86	0.77	0.42	-	-	0.09	0.63	0.77	0.78	0.74	0.56	0.00
		95% CI									95% CI							
Reliability	*r_xx_*	Inf.	Sup.							*r_xx_*	Inf.	Sup.						
*ω*	0.90	0.89	0.91							0.84	0.83	0.85						
*α*	0.90	0.89	0.91							0.84	0.83	0.85						

Note. F: unidimensional factor and factor loadings; *a*_irt_: IRT item discrimination; SEM: standard error of measurement; Stand. Inf.: standardized information function; *ω* and *α*: omega and alpha reliability coefficients; *r_xx_*: estimation of reliability.

**Table 4 ijerph-19-07978-t004:** Measurement invariance and correlation of the BSSS scores with sex and age.

	BSSS-8					BSSS-4				
	ULSMV-*χ^2^* (gl)	CFI	∆_CFI_	SRMR	∆_SRMR_	ULSMV-*χ^2^* (gl)	CFI	∆_CFI_	SRMR	∆_SRMR_
Sex										
Configuration	493.63 (40)	0.994	-	0.049	-	81.05 (4)	0.990	-	0.053	-
Thresholds	498.10 (56)	0.994	0.00	0.049	0.00	86.68 (12)	0.990	0.00	0.053	0.00
Loadings + thresholds	523.83 (63)	0.994	0.00	0.049	0.00	98.42 (15)	0.989	0.00	0.056	0.00
Intercepts	638.83 (70)	0.992	0.00	0.049	0.00	163.64 (18)	0.980	0.00	0.059	0.00
Residuals	685.12 (78)	0.992	0.00	0.051	0.00	190.76 (22)	0.977	0.00	0.055	0.00
Age										
Configuration	496.72 (40)	0.994	-	0.049	-	81.21 (4)	0.990	-	0.053	-
Thresholds	500.59 (56)	0.994	0.00	0.049	0.00	83.65 (12)	0.990	0.00	0.053	0.00
Loadings + thresholds	504.80 (63)	0.994	0.00	0.049	0.00	87.66 (15)	0.990	0.00	0.054	0.00
Intercepts	530.44 (70)	0.994	0.00	0.049	0.00	107.93 (18)	0.988	0.00	0.054	0.00
Residuals	552.67 (78)	0.994	0.00	0.050	0.00	117.31 (22)	0.987	0.00	0.054	0.00

Note. ∆_CFI_: differences in the CFI. ∆_SRMR_: differences in the SRMR. BSSS-8 and BSSS-4: full and short versions of the BSSS, respectively.

**Table 5 ijerph-19-07978-t005:** Monotonic and linear association between BSSS scores and depressive symptoms.

	BSSS-8		BSSS-4	
	Zero Order	Partial Correlation ^a^	Zero Order	Partial Correlation ^a^
CESD-7				
Linear	0.239 **	0.244 **	0.232 **	0.239 **
Monotonic	0.233 **	0.238 **	0.226 **	0.235 **
Descriptive statistics				
Mean	25.01	-	12.61	-
SD	8.62	-	4.59	-
Skew	−0.27	-	−0.21	-
Kurtosis	−0.75	-	−0.83	-
Distribution	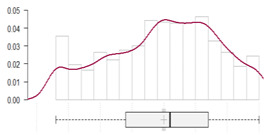		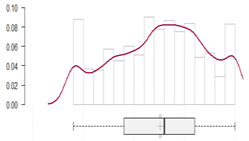	

Note. CESD-7: raw score of the Center for Epidemiological Studies Depression Scale-7. **^a^** partial correlation, adjusted for sex and age. ** *p* < 0.01.

**Table 6 ijerph-19-07978-t006:** Hierarchical logistic regression for incremental validity.

	BSSS-8				BSSS-4			
	B	Exp(B)	Variable Importance	McFadden *R*^2^	B	Exp(B)	Variable importance	McFadden *R*^2^
Alcohol consumption								
Step 1				0.039				0.039
Sex	0.108	1.114	1.38		0.108	1.114	1.38	
Age	0.116 **	1.123	3.98		0.116 **	1.123	3.98	
CESD	0.099 **	1.104	11.64		0.099 **	1.104	11.64	
Step 2				0.065				0.055
BSSS score	0.042 **	1.04	8.91		0.068 **	1.07	7.79	
Tobacco consumption								
Step 1				0.037				0.037
Sex	0.219 **	1.24	2.54		0.219 **	1.24	2.54	
Age	0.107 **	1.11	3.70		0.107 **	1.11	3.70	
CESD	0.097 **	1.10	10.46		0.097 **	1.10	10.46	
Step 2				0.058				
BSSS score	0.044 **	1.04	8.25		0.074 **	1.07	7.53	0.054

Note. BSSS: raw score of Brief Sensation Seeking Scale (8- and 4-item versions). Exp(B): equivalent odds ratio of the B coefficient. CESD-7: raw score of the Center for Epidemiological Studies Depression Scale-7. ** *p* < 0.01.

## Data Availability

The raw data supporting the conclusions of this article are available upon reasonable request.

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
