# Peer review of "Brief Sensations Seeking Scale (BSSS): Validity Evidence in Mexican Adolescents"

_ijerph, 2022, doi:10.3390/ijerph19137978_

Round 1

Reviewer 1 Report

Title
The title of the article concerns the adaptation Brief Sensation Seeking Scale in Mexican Adolescents. I think, the topic is worth researching. 

Abstract
The abstract is rather informative and it reflects the body of the paper. However, I think, authors should add information about additional characteristic of participants, ex.:  age (M; SD) or median and gender.

1.Introduction
The theoretical introduction contains a list of different theoretical positions concerning aspects of sensation seeking. The introduction is well arranged. The logic of the text is easy to follow. This part of manuscript provides sufficient background information. 
I suggest - you can add information about the limitations of the full-scale sensation-seeking (See: Arnett, 1994 or Hoyle, 2002).

2.Materials and Methods
2.3.1.Please add sentence:  The response format follows a 5-level Likert type scale 

3.Results
3.2. Item Analysis
You can add information about Mardia’s standardized estimator of the multivariate kurtosis 
3.2.1.You write that model was satisfactory. Please write what it means: You can add information about theoretically acceptable values for this indices (See: Hu & Bentler, 1999).
3.2. Measurement Invariance 
Should be: 3.2.1. Measurement Invariance 
In this paragraph : Please add information on why you split the group according to these criteria (14-16 years old and 17 years old and over groups)
3.3. Internal Consistency
Is: ………coefficients were the same for the BSSS-8 and BSSS-4 (see Table 3).
Should be: ….. coefficients were similar for the BSSS-8 and BSSS-4 (see Table 3).

4.Discussion section 
In  Discussion section conclusions are interesting  and include limitations of the study. 
You can articulate the directions of future research more strongly.
Congratulations!
Reviewer

Author Response

Responses to reviewers

Brief Sensations Seeking Scale (BSSS): Validity evidences in Mexican Adolescents

Prof. Dr. Paul B. Tchounwou

Editor-in-Chief

Thank you for the opportunity to resubmit our manuscript ID: ijerph-1683724 "Brief Sensations Seeking Scale (BSSS): Validity evidences in Mexican Adolescents" to International Journal of Environmental Research and Public Health (IJERPH). We appreciate the opportunity to publish in IJERPH. We have carefully reviewed the reviewers' valuable comments. To ensure that we have fully addressed all of your concerns, we have revised our manuscript based on the reviewers' suggestions.

Below, we include our point-by-point responses to the reviewers' corrections and comments and describe in detail the changes made to the manuscript.

Thank you again for the opportunity to resubmit our manuscript. I appreciate any help you can provide.

Sincerely,

Filiberto Toledano-Toledano, Ph.D.

Federico Gómez Children’s Hospital of Mexico, National Institute of Health.

Dr. Márquez 162, Doctores, Cuauhtémoc, México City, 06720, México.

+ 52 55 52289917, ext. 4318. E-mail: filiberto.toledano.phd@gmail.com

Review Report (Reviewer 1)

English language and style

( ) Extensive editing of English language and style required
( ) Moderate English changes required
( ) English language and style are fine/minor spell check required
( X) I don't feel qualified to judge about the English language and style

Comments and Suggestions for Authors

General comment

Title

The title of the article concerns the adaptation Brief Sensation Seeking Scale in Mexican Adolescents. I think, the topic is worth researching.

Response:

Thank you for pointing out your observation, and the constructive response that strengthens the manuscript.

Change:

None.

Abstract

The abstract is rather informative and it reflects the body of the paper. However, I think, authors should add information about additional characteristic of participants, ex.:  age (M; SD) or median and gender.

Response:

Thank you for pointing out your observation. We made the change.

Change:

“The sample consisted of 2,884 students (age: M= 16.6, SD = 1.5)…”

1.Introduction

The theoretical introduction contains a list of different theoretical positions concerning aspects of sensation seeking. The introduction is well arranged. The logic of the text is easy to follow. This part of manuscript provides sufficient background information.

I suggest - you can add information about the limitations of the full-scale sensation-seeking (See: Arnett, 1994 or Hoyle, 2002).

Response:

Thank you for this observation. We add information.

Change:

The SSS-V showed some problems that limited its use in the following years. These limitations were the extension with respect to the number of items, the unstable relationship of the items with their intended constructs to measure, the age of the content, and the reference to adult situations [26, 27]. The demand for a brief, adolescent-usable, unidimensional, and group-invariant measure motivated the development of a measure, the Brief Sensations Seeking Scale (BSSS; Hoyle et al. [26]). …“To date, there has been a comparative increase in the number of BSSS studies compared to SSS-V, and this trend is likely to continue for many years.”

2.Materials and Methods

2.3.1.Please add sentence:  The response format follows a 5-level Likert type scale

Response:

Thank you for this observation. We made the change.

Change:

“… with a 5-level points (from strongly disagree…”

3.Results

3.2. Item Analysis

You can add information about Mardia’s standardized estimator of the multivariate kurtosis

Response:

Thank you for this observation. We made the change.

Change:

“Both, Mardia’s skewness (b1p = 4.85, z = 2331.9, p < .01) and kurtosis (b2p = 88.99, z = 19.08, p < .01) tests shown non normality”

3.2.1.You write that model was satisfactory. Please write what it means: You can add information about theoretically acceptable values for this indices (See: Hu & Bentler, 1999).  

Response:

Thank you for pointing out your observation. We made the change.

Change:

“The fit was evaluated with two main approximate fit indices [58]: the comparative fit index (CFI; good fit, ≥ .95 [58]) and the standardized root mean squared residual (SRMR; good fit, ≤ .05 [58]) index. “

3.2. Measurement Invariance

Should be: 3.2.1. Measurement Invariance

In this paragraph : Please add information on why you split the group according to these criteria (14-16 years old and 17 years old and over groups)

Response:

Thank you for pointing out your observation. We add the reason.

Change:

“This partitioning in variable age was decided on a basis of convenience for the analyses, thus balancing the size of the samples compared.”

3.3. Internal Consistency

Is: ………coefficients were the same for the BSSS-8 and BSSS-4 (see Table 3).

Should be: ….. coefficients were similar for the BSSS-8 and BSSS-4 (see Table 3).

Response:

Thank you for pointing out your observation. We understand that it is difficult to assert equality if it refers to the absolute equality between the alpha and omega coefficients in each version of the BSSS. I imagine that this inequality will appear at the decimal level. So, we add clarification.

Change:

“… were the similar for the BSSS-8 and BSSS-4 (see Table 3), and for pragmatic use, they can seen like equals”

4.Discussion section

In  Discussion section conclusions are interesting  and include limitations of the study.

You can articulate the directions of future research more strongly.

Congratulations!

Response:

Thank you for pointing out your observation, and the constructive response that strengthens the manuscript.

Change:

“Within a psychometric research context of the BSSS, future studies can focus on cross-cultural studies through a complete analysis of measurement invariance, as well as meta-analytic research to synthesize item parameters such as factor loadings, intercepts and residuals. SEM meta-analysis (MASEM) is indicated as an analysis strategy to synthesize cumulative evidence from independent studies. To date, we can assume that there is a sufficient amount to meta-analyze the internal structure of the BSS.”

Reviewer 2 Report

Thank you for the opportunity to review the manuscript titled Brief Sensations Seeking Scale (BSSS): Validity evidences in 2 Mexican Adolescents.

This is an online survey targeting high school students. The study aimed to test psychometric properties and quality of an existing assessments among Mexican high school aged students: the shorter version of the BSSS consisting of 4 items and the BSSS-8 (consisting of 8 items), and in comparison to each other.   

This reported study is part of a larger study. A total of 2,884 students participated in this study. Findings demonstrate that both tools do measure the construct of sensation seeking (in line with their purpose), and both can be utilized in research.

In general – This paper presents impressive statistical methods testing psychometric properties of both these tools. Thus, this study describes thorough statistical analyses.                                                                              Line 82-84 "The brevity of the BSSS-4 suggests that it is optimal for mass screening evaluations as it does not require a substantial time investment to complete". Completing the 8 item versions takes ~ 90 seconds (A very short time). This is not considered a substantial time investment. Further, the completion time difference between the 8 and 4 item versions is not presented. The authors did not provide sufficient rational for requiring such an extensive effort, which certainly has been invested in this study: Why does this matter?

Moreover, the authors mention that this is a part of a larger study (research project HIM/2015/017/SSA.1207), I would warmly recommend to broaden this paper perspective.

 Specifically:

-Title – evidences please consider using the word evidence

-Line 37: needs to be re-written

-Line 123 – Inclusion criteria – Under the age of 18 parental consents are required. Though written later on in this manuscript, I suggest adding here: Written informed consent was obtained from parents of participating students. In addition, was age restricted?

-While in lines 248-249 the authors present the recruitment by semesters, lines 165-166 state that recruitment was performed during "April and May 2019".  

Author Response

Review Report (Reviewer 2)

English language and style

( ) Extensive editing of English language and style required
( ) Moderate English changes required
() English language and style are fine/minor spell check required
( X) I don't feel qualified to judge about the English language and style

Yes

Can be improved

Must be improved

Not applicable

Does the introduction provide sufficient background and include all relevant references?

( )

( x)

( )

( )

Is the research design appropriate?

( x)

( )

( )

( )

Are the methods adequately described?

( x)

( )

( )

( )

Are the results clearly presented?

( x)

( )

( )

( )

Are the conclusions supported by the results?

( X)

( )

( )

( )

Comments and Suggestions for Authors

 Thank you for the opportunity to review the manuscript titled Brief Sensations Seeking Scale (BSSS): Validity evidences in 2 Mexican Adolescents.

This is an online survey targeting high school students. The study aimed to test psychometric properties and quality of an existing assessments among Mexican high school aged students: the shorter version of the BSSS consisting of 4 items and the BSSS-8 (consisting of 8 items), and in comparison to each other.  

This reported study is part of a larger study. A total of 2,884 students participated in this study. Findings demonstrate that both tools do measure the construct of sensation seeking (in line with their purpose), and both can be utilized in research.

In general – This paper presents impressive statistical methods testing psychometric properties of both these tools. Thus, this study describes thorough statistical analyses.

Response:

Dear reviewer. I thank you for your comment and implicit appreciation of our work.

Change:

None.

Line 82-84 "The brevity of the BSSS-4 suggests that it is optimal for mass screening evaluations as it does not require a substantial time investment to complete". Completing the 8 item versions takes ~ 90 seconds (A very short time). This is not considered a substantial time investment. Further, the completion time difference between the 8 and 4 item versions is not presented. The authors did not provide sufficient rational for requiring such an extensive effort, which certainly has been invested in this study: Why does this matter?

Response:

Thank you for pointing out your observation. We add an argumentation in the revised manuscript.

Change:

“Although in an absolute view, the time to complete the BSSS-8 version does not differ substantially from the time for the BSSS-4, from a relative angle this difference may be an influential factor. Specifically, this time difference in the context of a long survey, and administered under moderately optimal conditions, may not be perceived as trivial. Additionally, in an online survey where direct monitoring may not be in the control of the researcher, the brevity of the survey will be maximized with few items but without losing validity.”

Moreover, the authors mention that this is a part of a larger study (research project HIM/2015/017/SSA.1207), I would warmly recommend to broaden this paper perspective.

Response:

Thank you for pointing out your observation. We add an argumentation in the revised manuscript.

Change:

This manuscript integrates methodological aspects of the global project.

Specifically:

-Title – evidences please consider using the word evidence

Response:

We made the change.

Change:

“Brief Sensations Seeking Scale (BSSS): Validity evidence in…”

-Line 37: needs to be re-written

Response:

Thank you for pointing out your observation. We made the change.

Change:

“Although initially sensation seeking (SS) was initially considered a biologically…”

-Line 123 – Inclusion criteria – Under the age of 18 parental consents are required. Though written later on in this manuscript, I suggest adding here: Written informed consent was obtained from parents of participating students. In addition, was age restricted?

Response:

Thank you for this observation. But we think that an equivalent statement already exists in our manuscript:

“First, the consent form was given to the parents or guardians so they could decide whether to allow their children to participate in the study.”

There was no age restriction for participation.

Change:

None.

-While in lines 248-249 the authors present the recruitment by semesters, lines 165-166 state that recruitment was performed during "April and May 2019".

Response:

Semesters are the number of semesters completed by students. We add the following clarification.

Change:

“…. (semesters are the number of semesters completed by students, and is an indicator of their progress in their studies.)”

Reviewer 3 Report

Thank you for the opportunity to review the manuscript entitled “Brief Sensations Seeking Scale (BSSS): Validity evidences in Mexican Adolescents.” This paper adds to the literature by providing additional evidence regarding the psychometric properties of the BSSS-8 and BSSS-4, which have important research implications. Overall, the manuscript is well-written and I have a few comments on ways to enhance the overall quality of the manuscript:

Introduction

  • Takes a while to get to the rationale for the current study. Condensing the background information (paragraphs 1-3) and more clearly laying out the gaps in the literature would strengthen the introduction.
  • More clearly stating the implications of our limited understanding certain psychometric properties on research and clinical practice would further enhance the rationale for the current study.

Methods

  • When describing the participants (section 2.2), please state that the participants are students. This is not clear until much later in the methods section.
  • Do the five cities/schools differ on potential confounding variables (e.g., academics, employment, income, health-related constructs)? If so, the authors might consider accounting for these site-level differences in their analyses.
  • It would be helpful to provide some more information about the variability with regards to participant age (i.e., IQR).
  • Please add the recruitment rates for each school.

Results

  • The results are well-written. I do not have any suggestions on ways to improve this section.

Discussion

  • The findings around the accuracy of the BSSS-4 and BSSS-8 I think are particularly important and have implications for studies on sensation-seeking and substance use. Are there any recommendations on what can be done to help address this limitation in studies where investigators are interested in high or low levels of sensation-seeking (e.g., alternative measures)?
  • In the introduction, the authors briefly mention the potential utility of the BSSS-4 and BSSS-8 with regards to mass screening. However, screening is only briefly mentioned in the discussion. A more thorough discussion of the clinical implications of this study (e.g., does the BSSS-4 possess any advantage to the BSSS-8) would enhance the overall quality of the discussion.

Author Response

Review Report (Reviewer 3)

English language and style

( ) Extensive editing of English language and style required
( ) Moderate English changes required
(x) English language and style are fine/minor spell check required
( ) I don't feel qualified to judge about the English language and style

Yes

Can be improved

Must be improved

Not applicable

Does the introduction provide sufficient background and include all relevant references?

( )

( X)

( )

( )

Are all the cited references relevant to the research?

(X)

Is the research design appropriate?

( X)

( )

( )

( )

Are the methods adequately described?

( )

( )

(X )

( )

Are the results clearly presented?

( )

(X )

( )

( )

Are the conclusions supported by the results?

(X )

( )

( )

( )

Comments and Suggestions for Authors

Thank you for the opportunity to review the manuscript entitled “Brief Sensations Seeking Scale (BSSS): Validity evidences in Mexican Adolescents.” This paper adds to the literature by providing additional evidence regarding the psychometric properties of the BSSS-8 and BSSS-4, which have important research implications. Overall, the manuscript is well-written and I have a few comments on ways to enhance the overall quality of the manuscript:

Response:

Thank you very much for the constructive responses that strengthens the manuscript.

Change:

None.

Introduction

   Takes a while to get to the rationale for the current study. Condensing the background information (paragraphs 1-3) and more clearly laying out the gaps in the literature would strengthen the introduction.

Response:

Thank you for pointing out your observation. In our manuscript, there is already a summary of the gaps in the literature on BSSS. Here is the paragraph:

“However, recent studies (presented in Table 1) suggest that several aspects of the BSSS are not addressed or inconsistently addressed in the analyses of its internal structure. These aspects include the absence of conditional reliability, equivalence of the full and short versions, treatment of items as continuous variables, psychometric equivalence between groups, and the fact that internal consistency is mainly estimated by the α coefficient. Furthermore, the psychometric value of the abbreviated 4-item version (the BSSS-4; [27]) remains unclear.”

….

And for each of these points, in the following paragraph we extend the explanations. We have tried to expand on the problems that our study has solved:

“In sum, recent studies have indicated that a) the conditional reliability of the BSSS may represent a realistic characteristic of score accuracy at different score levels [33], b) the equivalence of the two versions of the BSSS is required to guarantee the conceptual repre-sentativeness of a modified version of the instrument and establish consistency in deci-sions using the scores [34,35], c) treating categorical items as continuous variables pro-duces spurious and unrepresentative estimates of their statistical properties [36], d) the coefficient α is not efficient measure of internally consistent if there are correlated errors and wide-ranging differences in factor loadings [37], and e) measurement invariance is an aspect of internal structure that allows the use of the instrument to compare groups, such as with means or correlations, but it is infrequently assessed in BSSS [33].”

If the reviewer requires other information, please let us know.

Change:

None.

    More clearly stating the implications of our limited understanding certain psychometric properties on research and clinical practice would further enhance the rationale for the current study.

Response:

Thank you for this observation. We add to the manuscript relevant arguments on the clinical practice of BSSS use.

Change:

“Implications of clinical value can be concluded in our study. In this study, we extend the validity of the BSSS in a new population context (Mexico), but we also add other properties not addressed in previous studies that have potential clinical value. For example, item scaling performance is not only a matter of statistical item distribution, but is also potentially linked to item clinical validity, because the response options have clinical value when it helps differentiate individual risk factors. Second, conditional reliability helps to assess the accuracy of the scores at the construct levels where the interventions are applied, i.e., high levels.”

Methods

    When describing the participants (section 2.2), please state that the participants are students. This is not clear until much later in the methods section.

Response:

Thank you for this observation. We made the change.

Change:

“Students participants from five high schools…”

    Do the five cities/schools differ on potential confounding variables (e.g., academics, employment, income, health-related constructs)? If so, the authors might consider accounting for these site-level differences in their analyses.

Response:

Thank you for pointing out your observation. We had no additional data on explorer differences in academics, employment, income, health-related constructs, and other variables. But to highlight this limitation, we add in an explanatory paragraph in the limitations

Change:

“Finally, the possible variability of psychometric properties across schools was not explored, and this may require a multilevel approach. However, because sensation seeking is understood primarily as an individual attribute, it may be reasonable to assume that this variability is not substantial.”

    It would be helpful to provide some more information about the variability with regards to participant age (i.e., IQR).

Response:

Thank you for pointing out your observation. We made the change.

Change:

“… (median: 17 years old; IQR = 1)”

    Please add the recruitment rates for each school.

Response:

Thank you for pointing out your observation. We regret to report that recruitment rates for each school were not estimated, and we have no data available for this exact calculation. We avoid giving any answer in the manuscript to avoid this impression.

Change:

None.

Results

    The results are well-written. I do not have any suggestions on ways to improve this section.

Response:

Thank you very much for your appreciation of this section.

Change:

None

    The findings around the accuracy of the BSSS-4 and BSSS-8 I think are particularly important and have implications for studies on sensation-seeking and substance use. Are there any recommendations on what can be done to help address this limitation in studies where investigators are interested in high or low levels of sensation-seeking (e.g., alternative measures)?

Response:

Thank you for pointing out your observation. We have written implications of clinical value in the other answers, and we believe that these answers can also point to what is required by the revisor in this question. If additional explanations are required, please let us know.

Change:

“Implications of clinical value can be concluded in our study. In this study, we extend the validity of the BSSS in a new population context (Mexico), but we also add other properties not addressed in previous studies that have potential clinical value. For example, item scaling performance is not only a matter of statistical item distribution, but is also potentially linked to item clinical validity, because the response options have clinical value when it helps differentiate individual risk factors. Second, conditional reliability helps to assess the accuracy of the scores at the construct levels where the interventions are applied, i.e., high levels. Finally, an assessment of a number of individual variables associated with risk behaviors requires the school team to plan the selection of instruments and balance the costs of their administration. In this sense, the BSSS-4 can be an excellent tool for screening variables of clinical impact, especially when it comes to preventing substance use in adolescents. Due to its brevity and consistently satisfactory psychometric properties, the BSSS can be incorporated into universal screening of adolescents in school. Measures of depression may also be included in this screening, and possibly the instrument used here, the Center for Epidemiological Studies Depression Scale - 7 (CESD-7; [48]) may be of particular importance.”

    In the introduction, the authors briefly mention the potential utility of the BSSS-4 and BSSS-8 with regards to mass screening. However, screening is only briefly mentioned in the discussion. A more thorough discussion of the clinical implications of this study (e.g., does the BSSS-4 possess any advantage to the BSSS-8) would enhance the overall quality of the discussion.

Response:

Thank you for pointing out your observation. We made the change.

Change:

“An assessment of a number of individual variables associated with risk behaviors requires the school team to plan the selection of instruments and balance the costs of their administration. In this sense, the BSSS-4 can be an excellent tool for screening variables of clinical impact, especially when it comes to preventing substance use in adolescents. Due to its brevity and consistently satisfactory psychometric properties, the BSSS can be incorporated into universal screening of adolescents in school. Measures of depression may also be included in this screening, and possibly the instrument used here, the Center for Epidemiological Studies Depression Scale - 7 (CESD-7; [48]) may be of particular importance.”

Round 2

Reviewer 2 Report

The manuscript has been updated, yet I still have few concerns.

Added lines 95-101:  The authors  should base their reasoning in scientific publications, no references are presented.  

English editing is recommended (e.g., Although initially sensation seeking (SS) was initially…), and especially for the added parts.

Author Response

May 13, 2022, Mexico City

Responses to reviewers

Brief Sensations Seeking Scale (BSSS): Validity evidences in Mexican Adolescents

Prof. Dr. Paul B. Tchounwou

Editor-in-Chief

On behalf of the authors, I would like to thank you for the opportunity to resubmit our manuscript ID: ijerph-1683724"Brief Sensations Seeking Scale (BSSS): Validity evidences in Mexican Adolescents" to the International Journal of Environmental Research and Public Health (IJERPH). We appreciate the opportunity to publish in IJERPH. We have carefully reviewed the reviewers' valuable comments. To ensure that we have fully addressed all of your concerns, we have revised our manuscript based on the reviewers' suggestions.

Below, we include our point-by-point responses to the reviewers' corrections and comments and describe in detail the changes made to the manuscript.

Thank you again for the opportunity to resubmit our manuscript; furthermore, the authors of this study would appreciate, it if any additional observations arise from your final revision.

Sincerely,

Filiberto Toledano-Toledano, Ph.D.

Hospital Infantil de México Federico Gómez National Institute of Health.

Dr. Márquez 162, Doctores, Cuauhtémoc, México City, 06720, México.

+ 52 55 52289917, ext. 4318. E-mail: filiberto.toledano.phd@gmail.com

Review Report (Reviewer 2)

English language and style

( ) Extensive editing of English language and style required
( ) Moderate English changes required
() English language and style are fine/minor spell check required
( X) I don't feel qualified to judge about the English language and style

Yes

Can be improved

Must be improved

Not applicable

Does the introduction provide sufficient background and include all relevant references?

( )

( x)

( )

( )

Is the research design appropriate?

( x)

( )

( )

( )

Are the methods adequately described?

( x)

( )

( )

( )

Are the results clearly presented?

( x)

( )

( )

( )

Are the conclusions supported by the results?

( X)

( )

( )

( )

Comments and Suggestions for Authors

The manuscript has been updated, yet I still have few concerns.

Added lines 95-101:  The authors  should base their reasoning in scientific publications, no references are presented. 

English editing is recommended (e.g., Although initially sensation seeking (SS) was initially…), and especially for the added parts.

Response:

Dear reviewer. I thank you for your observation. Our reasoning is based on two things: first on our long experience as researchers, especially in the use of short scales, but also on the literature directly related to the use of the BSSS. For this reason, we put references derived from observations of previous BSSS research.

Also, we made an edition of the added parts.

Change:

“… sensation seeking measurement [26, 27, 31].”

We greatly appreciate all the contributions of the reviewers that we are sure will enrich our manuscript.

Sincerely,

Filiberto Toledano-Toledano, Ph.D.